# MOVEMENT-TO-ACTION TRANSFORMER NETWORKS FOR TEMPORAL ACTION PROPOSAL GENERATION

## ABSTRACT

The task of generating temporal action proposals is aimed at identifying temporal intervals containing human actions in untrimmed videos. For arbitrary actions, this requires learning long-range interactions. We propose an end-to-end Movement-to-Action Transformer Network (MatNet) that uses results of human movement studies to encode actions ranging from localized, atomic, body part movements, to longer-range, semantic movements involving subsets of body parts. In particular, we make direct use of the results of Laban Movement Analysis (LMA). We use LMA-based measures of movements as computational definitions of actions.From the input of RGB + Flow (I3D) features and 3D pose, we compute LMA based low-to-high-level movement features, and learn action proposals by applying two heads on the boundary Transformer, three heads on the proposal Transformer and using five types of losses. We visualize and explain relations between the movement descriptors and attention map of the action proposals. We report results from a number of experiments on the Thumos14, ActivityNet and PKU-MMD datasets, showing that MatNet achieves SOTA or better performance on the temporal action proposal generation task. [1]

## 1 INTRODUCTION

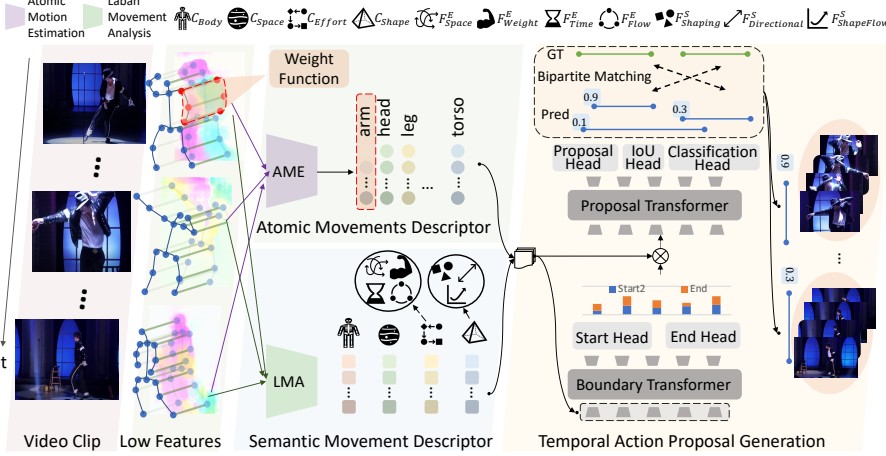

Figure 1: Overview of our MatNet architecture. It contains two main components: (1) Movement descriptors, shown in the middle column, and (2) Transformer networks for action proposal generation, shown in the right column. Given a sequence of untrimmed video frames, MatNet uses Laban Movement Analysis constructs to generate body part (atomic) level and subset-of-parts (semantic) level descriptors of human movements from videos, and input as movement representations to action-boundary sensitive Transformer networks to generate action proposals.

With advances in the understanding of trimmed, human action videos, the focus has begun to shift to longer and untrimmed videos. This has increased the need for segmentation of the videos into action clips, namely, identification of temporal intervals containing actions. This is the goal of the task of temporal action proposals generation (TAPG) for human action understanding. There are many factors that make the problem challenging: (1) Location: an action can start at any time. (2) Duration: The time taken by the action can vary greatly. (3) Background: Irrelevant content can

---

[1] We will make all of the data sets, resources and programs publicly available.

| Comp. | Description | Factor | Variable | | Cognition and Geometry | |
|---|---|---|---|---|---|---|
| **1. Non-Kinematic** — Effort | Movement dynamics: strength, control, timing to reflect inner intention | Space | $c_1 \in \mathbb{R}^{T\times 2}$ | *Cognitive* | Attention | {Direct,Indirect} |
| | | Weight | $c_2 \in \mathbb{R}^{T\times 1}$ | | Intention | {Strong,Light} |
| | | Time | $c_3 \in \mathbb{R}^{T\times 1}$ | | Decision | {Sudden,Sustained} |
| | | Flow | $c_4 \in \mathbb{R}^{T\times 1}$ | | Progression | {Free,Bound} |
| Shape | Shape change during movement | Shaping | $c_5 \in \mathbb{R}^{T\times 3}$ | *Plane* | Vertical / Horizontal / Sagittal | {Rising,Sinking} / {Spreading,Enclosing} / {Advancing,Retreating} |
| | | Directional | $c_6 \in \mathbb{R}^{T\times 13}$ | | – – | {Spoke-like,Arc-like} |
| | | Shape Flow | $c_7 \in \mathbb{R}^{T\times 1}$ | | – – | {Growing,Shrinking} |
| **2. Kinematic** — Body | Structural & physical properties of moving body | Body | $c_8 \in \mathbb{R}^{T\times 14}$ | | – – | – |
| Space | Movement patterns, trajectories & tension | – | – | | – – | – |
| 3. Relationships | Interaction among people, body parts & objects | – | – | | – – | – |

Table 1: LMA classifies movement into three main categories (Col 1) Santos (2014): (1) Non-Kinematic (2) Kinematic, each represented by two Components (Col 2) - Effort and Shape for (1), and Body and Space for (2). Category 3 is about Relationships between (1) and (2) (Col 1, 2). The components describe the categories using eight Factors, denoted by multidimensional variables $\{c_i\}_{i=1}^{8}$, (Col 4), each having a different dimension (Col 5). Each Factor is also associated with a movement's underlying (cognitive) category, and its geometric structure and space (plane), that capture the position, direction, rotation, velocity, acceleration, distance, curvature and volume associated with the movement, and have values between two extremes (Col 7). Human movement analysts point out that although each individual may combine the Factors in ways that are specific to the individual and their cultural, personal and artistic preferences, $\{c_i\}_{i=1}^{8}$ remain valid for all movements Bartenieff & Lewis (1980) and human activities Santos (2014), and can be used to describe human movement at the semantic level. In this paper, we use these Factors as bases to obtain temporal action proposals through MatNet. MatNet automatically determines combinations of the Factors most suited for action detection and localization.

be highly diverse. (4) Number of Actions: Unknown and unlimited. (5) Action Set: Unknown (6) Ordering: Unknown. Many problems can benefit from accurate localization of human activities, such as activity recognition, video captioning and action retrieval Ryoo et al. (2020); Deng et al. (2021). Recent approaches can be divided into top-down (anchor-based) Gao et al. (2017a); Liu et al. (2019a); Gao et al. (2020) and bottom-up (boundary-based) Lin et al. (2019); Su et al. (2021); Islam et al. (2021). The former employ a fixed-size sliding window or anchors to first predict action proposals, and then refine their boundaries based on the estimated confidence scores of the proposals. The latter first generate probabilities that each frame is in the middle or at a boundary of an action, and then obtain an optimal proposal based on the confidence scores of the proposals. However, the confidence scores are based on local information, and without making full use of long-range (global) context. Although different techniques have been proposed to model local and global contextual information, the video information they use is low-level. They do not incorporate multilevel representations, e.g., from low-level video features to higher-level models of human body structure and dynamics. In this paper, we incorporate such knowledge using Laban theory of human movement Guest (2005).

Laban Movement Analysis (LMA) is a widely used framework that captures qualitative aspects of movement important for expression and communication of actions, emotions, etc. Originally, LMA characterizes movement using five components: Body, Effort, Shape, Space and Relationship (Table 1). Each addresses specific properties of movement and can be represented in Labanotation Guest (2005), a movement notation tool. LMA is a good representation, integrating high-level semantic features and low-level kinematic features. Li et al. (2019) analyzes different kinds of dance movements and generates Labanotation scores. However, they work with manually trimmed videos. There are some early works that segment dance movements using LMA. Bouchard & Badler (2007) detects movement boundaries from large changes in a series of weighted LMA components. SONODA et al. (2008) presents a method of segmenting whole body dance movement in terms of "unit movements" which are defined using LMA components of effort, space and shape; constructed based on the judgments of dance novices, and used as primitives. However, they do not use hierarchical

motion patterns, which are central to human description of dances (e.g., patterns like hop-step left-hop-step right), and are therefore limited to only the basic movements.

In this paper, we propose end-to-end Movement-to-Action Transformer Networks (MatNet) for temporal action proposal generation. As shown in Figure 1, our TAPG Transformer consists of two main types of modules: Movement Descriptors and Transformer networks. The former includes an atomic movement descriptor $F_a$ (Sec. 4.1), that can recognize movements of each body part, and a semantic movement descriptor $F_s$ (Sec. 4.2), that can quantitatively describe the human movement. The Transformer networks are comprised of a boundary Transformer $\Phi_b$ (Sec. 4.3) and a proposal Transformer $\Phi_p$ (Sec. 4.4); they enable capturing long-range contextual information.

The main contributions of this paper are as follows:

- We propose end-to-end Movement-to-Action Transformer Networks that use a range of low (atomic) to high (semantic) level human movements for temporal action proposal generation.

- Our high level features are based on movement concepts evolved by human movement (e.g., dance) experts.

- Our method is robust to occlusions of humans by other humans or objects because our underlying human pose detector (using LCRNet) has such robustness.

- Our boundary and proposal Transformers can be jointly trained end-to-end on any new dataset, without requiring a pretrained backbone network.

- Experiments show that the results of the proposed MatNet are superior to those of the state-of-the-art methods on the PKU-MMD Chunhui et al. (2017) and Thumos14 Jiang et al. (2014) datasets, and are comparable on ActivityNet Fabian Caba Heilbron & Niebles (2015). Although estimation of the 3D pose is a challenging task, our results demonstrate that including it in the input significantly improves the results.

## 2 RELATED WORK

**Temporal Action Segmentation** aims to segment an untrimmed video and label each segment with one of a set of pre-defined action labels. Some network-based methods capture both short- and long-term dependencies Farha & Gall (2019); Gao et al. (2021). Some methods used are weakly-supervised Li et al. (2021); Fayyaz & Gall (2020). Others propose unsupervised segmentation of complex activities without any additional input Sarfraz et al. (2021); Li & Todorovic (2021).

**Temporal Action Proposal Generation**. Unlike temporal action segmentation, which refers to both action localization and recognition of actions from a given set, temporal action proposal generation (TAPG) is about the more general localization of actions without knowledge of the names, numbers, and order of the actions, which requires long-range global information. Many methods have been proposed to model local and global contextual information in videos Chéron et al. (2015); Gu et al. (2018); Zolfaghari et al. (2017); Choutas et al. (2018); Zhang et al. (2018); Asghari-Esfeden et al. (2020); Hsieh et al. (2022); Qing et al. (2021a). In addition, Xu et al. (2020); Chen et al. (2021) formulate the action detection problem as a sub-graph localization problem using a graph convolutional network (GCN). By providing more flexible and precise action proposals, TAPG can help to correct the results of the temporal action segmentation or action recognition and provide a foundation for other applications.

**Transformer Based Methods**. RTD-Net Tan et al. (2021) uses Transformer for proposal generation, by weighing the input of the proposal Transformer by pre-calculated boundary scores. TAPGT Wang et al. (2021) proposes to adopt two Transformers to generate the boundary and proposal in parallel. TadTR Liu et al. (2022b) uses Transformer to map a small set of learned action query embeddings to corresponding action predictions adaptively with a Transformer encoder-decoder architecture. E2E-TAD Liu et al. (2022a) attaches a detection head to the last layer of the Transformer encoder, and optimizes the head and the video encoder simultaneously. ActionFormer Zhang et al. (2022) combines a multi-scale feature representation with local self-attention, and uses a light-weighted decoder to classify every moment in time and estimate the corresponding action boundaries. Our MatNet integrates the boundary-attention strategy from Wang et al. (2021) and the boundary and proposal Transformers from Tan et al. (2021), by directly multiplying the output of the boundary Transformers $\Phi_b$ with input features of the proposal Transformer $\Phi_p$, which, together with our movement descriptors, provides an end-to-end Movement-to-Action architecture with better modeling of long-and-short-term dependencies.

## 3 LMA AS A MOVEMENT DESCRIPTOR

Given an untrimmed video $\{I_t\}_{t=1}^T$, our goal is to generate a set of $N_g$ proposals $\hat{\Psi} = \{(\hat{t}_s^n, \hat{t}_e^n)\}_{n=1}^N$ (where $t_s$ and $t_e$ are the starting and ending times), that are close to the ground-truth action proposals $\Psi = \{(t_s^n, t_e^n)\}_{n=1}^N$. As in Fig. 1, proposed MatNet uses a range of features to capture the movements of body parts (limbs, joints, torso, head). The most primitive of these are their individual instantaneous displacements and temporal trajectories, which we call atomic movements, and combinations of atomic movements, as high level constructs we call semantic movements. While the atomic movements are basic primitives, for semantic features we do not use our own constructs; instead, we use descriptors evolved by dance experts. These descriptors use dance vocabulary given by the Laban Motion Analysis (LMA) Santos (2014) system, introduced in Santos (2014), and well studied and defined in terms of kinematics and dynamics equations. Table 1 presents details of the LMA representation. In the rest of this section, we present a brief overview of the five LMA components (Table 1, Col 2) that are central to our proposed methods.

(1) **Non-Kinemetic-Effort** captures dynamic characteristics with respect to "inner intention". It involves four factors: **Space -** $c_1$ describes the person's attention to the environment when moving, with values ranging from direct (single-focused) to indirect (multi-focused). It can be formulated as the moving direction $(\theta_x, \theta_y)$ of the person in the horizontal plane; a stable direction represents a direct movement and an unstable direction represents the opposite. **Weight -** $c_2$ describes the strength of the movement with intention on the person's own body, with values ranging from strong (fast and powerful) to light (slow and fragile). It can be formulated as the average velocity of body joints; a larger velocity represents a stronger movement. The **Time -** $c_3$ indicates if the person has decided and knows the right moment to move, ranging from sustained (leisurely) to sudden (instantaneous, in a hurry). It can be formulated as the average acceleration of body joints; a larger acceleration represents a more sudden movement. The **Flow -** $c_4$ describes control of the movement ranging from free (uncontrollable) to bound (controlled). It can be formulated as the sum of the jerks of the body joints; a larger jerk represents a freer movement.

(2) **Non-Kinemetic-Shape** studies the way the body changes shape during movement and involves three factors: **Shaping -** $c_5$ describes shape changes with respect to the environment as seen from the (x, y, z) directions, or projections on the vertical, horizontal and sagittal planes. It can be formulated as the area of convex hull of the body joint locations projected on the three planes; a large area on each plane represents rising, spreading and advancing, whereas a small area represents sinking, enclosing and retreating. **Directional -** $c_6$ again describes shape changes but in terms of joint movements, ranging from spoke-like (body joint moves in a direct line) to arc-like (body joint moves in an arc). It can be formulated as the curvature of the movement trajectory of each limb's end joint with root joint as the origin; a large curvature represents arc-like movement and a small curvature represents a more spoke-like movement. **Shape Flow -** $c_7$ describes self-motivated growing and shrinking of the "internal Kinesphere". It can be formulated as the volume of convex hull of 3D body joint locations; a large volume represents growing Kinesphere.

(3) **Kinemetic-Body** studies structural interrelationships within the body while moving, describing which body parts are moving, connected and influenced by others. It has only one factor, **Body -** $c_8$, which can be formulated as average rotation angles of joints.

(4) **Kinemetic-Space** is about the level and direction of a body part's movement. We skip this component since it is already captured by our low-level "atomic" body part movements and we want to capture only non-local, semantic movements.

(5) **Relationships** between a person and their surroundings. We also skip this component since it is less well defined and our current architecture focuses on the movement without modeling the surround and interactions with it.

## 4 PROPOSED APPROACH

A central theme of our approach is to represent movement in terms of the aforementioned eight, quantitative, semantic descriptors, $\mathbf{c} = \{c_i\}_{i=1}^8$, shown as different icons in the blue block (Semantic Movement Descriptor) in Fig 1 and derivable from mover's position, direction, rotation, velocity, acceleration, distance, curvature and volume. As shown in Fig. 1, our LMA based MatNet is composed of four main parts: an atomic movement descriptor $F_a$, a semantic movement descriptor $F_s$, a boundary transformer $\Phi_b$ and the final, proposal transformer $\Phi_p$. Since these parts depend on 3D pose $\mathbf{P} = \{P\}_{t=0}^{T-1}$, we first extract pose from the given video using LCRNet Rogez et al. (2019) which

detects multi-person 3D poses in natural images. To capture motion, we use the I3D representation of each frame $\mathbf{x} = \{x\}_{t=0}^{T-1}$ following Carreira & Zisserman (2017). Using these 3D pose and motion results, we implement our methods for these four parts. The atomic movement descriptor $F_a$ estimates the movements $\mathbf{m} = \{\{m_t^e\}_{e \in E}\}_{t=0}^{T-1}$ of each body part $e \in E$ from the trajectories $\{\{P_t^j\}_{j \in J_e}\}_{t=0}^{T-1}$ of all joints $j \in J_e$ connected to $e$, $J_e \subset E$. The semantic movement descriptor $F_s$ outputs kinematics-invariant LMA representations $\mathbf{c} = \{c\}_{t=0}^{T-1}$ of the entire body. Then the sequence of extracted movement features $\mathbf{m}$ and $\mathbf{c}$, together with 3D pose $\mathbf{P} = \{P\}_{t=0}^{T-1}$ and I3D features $\mathbf{x} = \{x\}_{t=0}^{T-1}$, are concatenated, denoted as $\mathbf{h} = \mathbf{m}^\frown \mathbf{c}^\frown \mathbf{P}^\frown \mathbf{x}$, and taken as input to the boundary transformer $\Phi_b$. The boundary transformer $\Phi_b$ generates the start and end boundary probabilities $\{(p_s^t, p_e^t)\}_{t=1}^{T}$ to weigh the input feature sequence $\mathbf{h}$, and the weighted feature sequence $\tilde{\mathbf{h}}$ is taken as input to the proposal transformer $\Phi_p$ to generate the action proposals $\hat{\Psi} = \{(\hat{t}_s^n, \hat{t}_e^n)\}_{n=1}^{N_g}$. The following subsections present details of the four parts.

## 4.1 Atomic Movement Descriptor

Since we do not have ground truth for many of the available large human action datasets, we use unsupervised methods for body part movement recognition, e.g., in Hu & Ahuja (2021). Given a sequence of 3D poses $\{\{\hat{p}_t^j\}_{j \in J_e}\}_{t=0}^{T-1}$ of all the joints $j \in J_e$ connected to a body part $e$, we classify the body part $e$'s movements $\mathbf{m}^e = \{\hat{m}_t^e\}_{t=0}^{T-1}$ using movement labels (left, right, etc.), and characterizing homogeneity of motion direction and level as in Hu & Ahuja (2021), representing all limb movements in a coordinate frame centered on torso. We first calculate the velocity $v^e$ of the end joint $j$ of each limb $e$, e.g., wrist for lower arm, elbow for upper arm, ankle for lower leg and knee for the upper leg, as follows:

$$\mathbf{v}_t^{ij} = \overrightarrow{P_t^i P_t^j} - \overrightarrow{P_{t-1}^i P_{t-1}^j}, \tag{1}$$

where $i$ and $j$ are the root and end joints of the limb, respectively. Then the velocity vector $\mathbf{v}_t^{ij}$ is transformed to be in the torso coordinate system:

$$\tilde{\mathbf{v}}_t^{ij} = \frac{\mathbf{v}_t^{ij} \cdot \mathbf{v}_t^{torso}}{||\mathbf{v}_t^{torso}||}, \tag{2}$$

where $\mathbf{v}_t^{torso}$ are the 3D coordinates of the torso. To extract major sustained movements in a direction, we identify salient peaks and valleys in the velocity profile $\Lambda = \{\tilde{\mathbf{v}}_t^{jk}\}_{t=1}^{T}$ curve. To smooth out the profile noise, we first smooth the velocity profile $\Lambda$ using a low pass filter while retaining a majority of the power. Then we identify $k$ peaks and valleys with the largest area. Finally, we save the timestamps of the $k$ peaks and valleys with the labels of the corresponding movements. In experiments, we estimate the movements $\mathbf{m}^e = \{\hat{m}_t^e\}_{t=0}^{T-1} \in \mathbb{R}^{T \times 45}$ of 14 body parts (arm, leg, torso, hip, shoulder, head), each having $2 \sim 4$ binary movement labels (e.g., move up vs move down, extension vs flexion).

## 4.2 Semantic Movement Descriptor

We now discuss how we compute the eight factors $\{c_i\}_{i=1}^{8}$ (Tab. 1).

**(1) $c_1$ - Space Effort:** $c_1$ ranges from direct (moving straight to the target) to indirect (not moving straight) Cui et al. (2019). Considering that the person is mostly moving in a horizontal plane, $c_1$ at time $t$ is defined as the heading direction in the x-y plane:

$$\text{Space:} \quad c_1 = [\theta x_t, \theta y_t]^T \tag{3}$$

**(2) $c_2$ - Weight Effort:** $c_2$ ranges from strong to light, and is estimated from the sum of the kinetic energy of the torso and distal body limbs (e.g., head, hands, feet). The higher the peak kinetic energy, the stronger the *Weight*. $c_2$ at time $t$ is defined as:

$$\text{Weight:} \quad c_2 = \sum_{j \in J} \alpha_j E^j(t) = \sum_{j \in J} \alpha_j v^j(t)^2 \tag{4}$$

where $J$ is the set of body joints, $\alpha_j$ is the mass coefficient for each joint, and $v^j(t)^2$ is the square of the speed of the joint at time $t$. since $c_2$ of a body joint is influenced mainly by its speed Samadani et al. (2020), we set the mass coefficients to 1 for all the body joints as in Hachimura et al. (2005); Samadani et al. (2020).

**(3) $c_3$ - Time Effort:** $c_3$ ranges from sudden to sustained. Sudden movements are characterized by large values in the acceleration sequence, as defined below, compared to sustained movements characterized by 0 acceleration. The acceleration for the $j^{th}$ body part at time $t$ is defined as the

change in velocity per unit time:

$$a^j(t) = \left| v^j(t) - v^j(t-1) \right|,\tag{5}$$

In Equation 5, acceleration is defined as the change in velocity over a unit time ($\Delta t = 1$). The sum of the accelerations of the torso and end-effectors is used to estimate $c_3$ for full-body movements:

$$\text{Time:}\quad c_3 = \sum_{j \in J} a^j(t).\tag{6}$$

**(4) $c_4$ - Flow Effort:** $c_4$ ranges from free to bound, and is computed as the aggregated jerk, third order derivative of the position, over a given time period $\Delta t$ (1 in our case) for the torso and end-effectors.

$$\text{Flow:}\quad c_4 = \sum_{j \in J} \left| a^j(t) - a^j(t - \Delta t) \right|\tag{7}$$

where $a^j(t)$ is the Cartesian acceleration of the $j^{th}$ body part at time $t$.

**(5) $c_5$ - Shape Shaping:** $c_5$ is used to primarily describe concavity and convexity of the torso in the (i) vertical, (ii) horizontal, and (iii) sagittal planes Lamb (1965), capturing Rising/Sinking (vertical plane), Widening/Narrowing (Horizontal plane), and Advancing/Retreating (Sagittal plane) Lamb (1965). (i) is due to the torso's upward-downward displacement Dell (1977), and quantified by its maximum value. (ii) is due to torso's forward-backward displacement Dell (1977), and is quantified by its maximum value. (iii) is mainly sideward over the body. As in Dell (1977), we estimate $c_5$ as the area of the convex hull of body's projection on the horizontal plane.

**(6) $c_6$ - Shape Directional:** $c_6$ ranges from spoke-like to arc-like, describes the transverse behavior of the limb movements Dell (1977), and is captured as the curvature of the movement of the end joint of the limb in a 2D plane within which the largest displacement of the limb occurs. we estimate it using the 2D curvature within the extracted 2D ($xy$ plane) at time $t$, as follows,

$$\text{Directional:}\quad c_6 = \frac{\sqrt{(\ddot{y}(t)\dot{x}(t) - \ddot{x}(t)\dot{y}(t))^2}}{(\dot{x}^2(t) + \dot{y}^2(t))^{3/2}}\tag{8}$$

where $\dot{x}(t)$ and $\ddot{x}(t)$ indicate the first and second derivatives of the $x$ trajectory at time $t$, respectively.

**(7) $c_7$ - Shape Flow:** $c_7$ ranges from growing to shrinking. Dell (1977) suggests the use of the "reach space" for estimating $c_7$. Three areas of reach are: 1) near (knitting), 2) intermediate (gesturing), and 3) far (space reached by the whole arm when extended out of the body without locomotion). Therefore, the limits of far reach are the limits of LMA's personal kinesphere, the space around the body which can be reached without taking a step Dell (1977). We estimate $c_7$ as the maximum volume of the convex hull (bounding box) containing the stretched body and limbs.

**(8) $c_8$ - Body:** $c_8$ captures the bending of joints throughout the body. For the example of arm, the bending degree ($c_8$) is calculated from the shoulder (S), elbow (E), and the wrist (W) joints, in terms of the two limb vectors $\overrightarrow{ES}$ and $\overrightarrow{EW}$, as follows:

$$c_8 = \arccos\left( \frac{\overrightarrow{ES} \cdot \overrightarrow{EW}}{|\overrightarrow{ES}||\overrightarrow{EW}|} \right)\tag{9}$$

### 4.3 BOUNDARY TRANSFORMER

The standard Transformer model is composed of an encoder and a decoder, with several feed-forward and multi-head layers. The multi-head self-attention layer models the interactions between the current frame and all other frames of a video sequence.

To keep the low-level information of the video, we extract the I3D (Inflated 3D Networks) Carreira & Zisserman (2017) representation as an additional feature. I3D is a widely adopted 3D convolutional network trained on the Kinetics dataset, and the I3D representation contains spatiotemporal information directly from videos. Then the I3D representations of all frames $\mathbf{x} = \{x\}_{t=0}^{T-1} \in \mathbb{R}^{T \times 2048}$ by Carreira & Zisserman (2017) are stacked with the 3D poses $\mathbf{P}$, and the two semantic movement representations $\mathbf{m}$ and $\mathbf{c}$, denoted as $\mathbf{h} \in \mathbb{R}^{T \times d}$, and given as the input to the boundary transformer $\Phi_b$ as well as to the proposal transformer $\Phi_p$.

To encode the proposal-level long-term temporal dependency for boundary regression, and then the frame-level short-term dependency for proposal generation, a Boundary Transformer $\Phi_b(\mathbf{h})$ is used to generate the starting and ending boundary probability sequences $(\mathbf{p}_s, \mathbf{p}_e) = \{(p_s^t, p_e^t)\}_{t=1}^{T} \in \mathbb{R}^{T \times 2}$, which are further multiplied with $\mathbf{h}$ as the weighted input to $\Phi_p$. We construct $\Phi_b$ using the standard

Transformer architecture described above. The encoder of $\Phi_b$ maps $\mathbf{h}$ stacked with a positional encoding Vaswani et al. (2017) to a hidden representation $\mathbf{h}_{enc} \in \mathbb{R}^{T \times d}$. The decoder of $\Phi_b$ takes the hidden representation $\mathbf{h}_{enc}^b$ as the query $Q$ and key $K$, and takes a value $V \in \mathbb{R}^{d_b \times d}$ initialized with zeros, and outputs a global representation of the boundaries $\mathbf{h}_{dec}^b \in \mathbb{R}^{d_b \times d}$ where $d$ is the feature vector size and $d_b$ is the number of queries. In $\Phi_b$, $d_b$ is the length of the sequence. Eventually, a starting boundary head and an ending boundary head, consisting of a multi-layer perceptron and a Sigmoid layer, are appended to generate the starting and ending probabilities $(\mathbf{p}_s, \mathbf{p}_e) \in \mathbb{R}^{T \times 2}$.

The **Loss function of the boundary transformer** is defined as:

$$\mathcal{L}_b = -\frac{1}{N_T} \sum_{t=1}^{N_T} \left( p_s^t \log y_s^t + \left(1 - p_s^t\right) \log \left(1 - y_s^t\right) \right)$$

$$- \frac{1}{N_T} \sum_{t=1}^{N_T} \left( p_e^t \log y_e^t + \left(1 - p_e^t\right) \log \left(1 - y_e^t\right) \right), \tag{10}$$

where $y_s^t$ and $y_e^t$ are the ground truth labels of the boundary.

### 4.4 Proposal Transformer

Unlike Tan et al. (2021), which uses an additional backbone network to generate and save the boundary scores which are un-trainable when training the proposal transformer, we multiply the starting and ending probabilities $(\mathbf{p}_s, \mathbf{p}_e)$ from the boundary transformer $\Phi_b$ with the input feature $\mathbf{h}$ to generate boundary-attentive representations $\tilde{h}$ as input to the proposal transformer $\Phi_p$. Similar to the architecture of boundary transformer $\Phi_b$, proposal transformer $\Phi_p$ takes the product of the boundary-attentive representations $\tilde{h}$ and a set of proposal queries $\in \mathbb{R}^{N_g \times d}$ as input, and outputs the proposal representations $\mathbf{h}_{dec}^p \in \mathbb{R}^{N_g \times d}$, where $N_g$ is the number of proposal queries, and the proposal queries are themselves a product of the training, initialized randomly. Eventually, a proposal head, a classification head and an IoU head are appended to generate a set of proposals $\hat{\Psi} = \{(\hat{t}_s^n, \hat{t}_e^n)\}_{n=1}^{N_g}$ that are close to the ground-truth action proposals $\Psi = \{(t_s^n, t_e^n)\}_{n=1}^{N}$, a set of proposal classification scores $\{p_{cls}^n\}_{n=1}^{N_g}$, and predicted IoUs $\{p_{iou}^n\}_{n=1}^{N}$.

The **Loss function of the proposal transformer** consists of a binary classification loss defined as:

$$\mathcal{L}_{cls} = -\frac{1}{N_g} \sum_{n=1}^{N_g} \left( p_{cls}^n \log y_{cls}^n + \left(1 - p_{cls}^n\right) \log \left(1 - y_{cls}^n\right) \right) \tag{11}$$

where $y_{cls}^n$ denotes the ground truth labels of proposal classification. An L1 regression loss to refine the boundaries is defined as:

$$\mathcal{L}_{loc} = \sum_{n=1}^{N_g} |\hat{t}_s^n - t_s^n| + |\hat{t}_e^n - t_e^n|; \tag{12}$$

a tIoU loss to measure the overlap is defined as:

$$\mathcal{L}_{tIoU} = \sum_{n=1}^{N_g} 1 - t\,\mathrm{IoU}\left(\psi^n, \hat{\psi}^n\right); \tag{13}$$

and an IoU prediction loss, as in Tan et al. (2021), is defined as:

$$\mathcal{L}_{IoU} = \sum_{n=1}^{N_g} ||p_{iou}^n - y_{iou}^n||_2^2 \tag{14}$$

### 4.5 Overall Objective Function

The overall objective function of the proposed MatNet is defined as a weighted summation of the boundary and proposal transformer losses in Sec 4.3 and 4.4:

$$\mathcal{L} = \alpha \mathcal{L}_b + \beta \mathcal{L}_{cls} + \lambda \mathcal{L}_{loc} + \omega \mathcal{L}_{tIoU} + \iota \mathcal{L}_{IoU}. \tag{15}$$

where we choose the weight values experimentally. We follow Tan et al. (2021) to iteratively train the different heads.

## 5 Experiments

We use the PKU-MMD Chunhui et al. (2017), ActivityNet Fabian Caba Heilbron & Niebles (2015) and THUMOS14 Jiang et al. (2014) for evaluation. **THUMOS14** has 413 temporally untrimmed videos. **ActivityNet** contains about 200 activity classes, with 10k training videos, 5k validation videos and 5k test videos. **PKU-MMD** contains 1076 videos of 51 action categories.

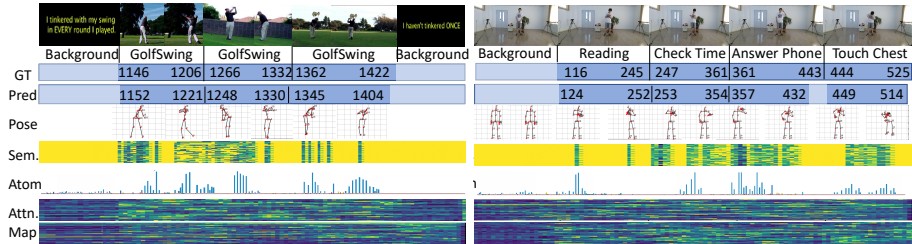

Figure 2: Qualitative visualization of the generated proposals (row 2) with corresponding poses (row 3), semantic movement features (row 4), atomic movement features (row 5), and attention map of the starting (row 6) and ending (row 7) boundaries on two samples from THUMOS14 Jiang et al. (2014) and PKU-MMD Chunhui et al. (2017) datasets.

We use Adam optimizer to train MatNet for 50 epochs. The learning rate is $1e - 4$, and the batch size is 32. We use 3 encoding layers and 6 decoding layers for both boundary and proposal Transformers. The length of the sequence $d_b$ is set to be 100 and the number of proposals $N_g$ expected to be generated is 32. We use a step size of 8 to extract frame sequences. We set the weights $\alpha, \beta, \lambda, \omega$ and $\iota$ in Eq. 4 to be 1, 1, 5, 2 and 100.

## 5.1 EVALUATION METRICS

We use the metric **AR@AN** (Average Recall (AR) over average number of proposals) under specified temporal Intersection over Union (tIoU) thresholds, which are set to [0.5: 0.05: 1] for Thumos14 and PKU-MMD, and [0.5: 0.05: 0.95] for ActivityNet. We also use the metric **mAP** (mean Average Precision under multiple tIoU). When a predicted temporal segment satisfies a tIoU threshold with the ground truth action label, this segment is considered as a true positive. The tIoU thresholds are set as 0.5, 0.75, 0.95 for ActivityNet, 0.3, 0.4, 0.5, 0.6, 0.7 for THUMOS-14 and 0.1, 0.3, 0.5 for PKU-MMD. We use the classifier of UntrimmedNet Wang et al. (2017) to compute the mAP scores.

## 5.2 QUANTITATIVE RESULTS

| Method | AR@AN | | | mAP | | | | |
|---|---|---|---|---|---|---|---|---|
| | @50 | @100 | @200 | 0.3 | 0.4 | 0.5 | 0.6 | 0.7 |
| TURN Gao et al. (2017b) ICCV | 21.9 | 31.9 | 43.0 | 46.3 | 35.3 | 24.5 | 14.1 | 6.3 |
| BSN Lin et al. (2018) ECCV | 37.5 | 46.1 | 53.2 | 53.5 | 45.0 | 36.9 | 28.4 | 20.0 |
| MGG Liu et al. (2019b) CVPR | 39.9 | 47.8 | 54.7 | 53.9 | 46.8 | 37.4 | 29.5 | 21.3 |
| BMN Lin et al. (2019) ICCV | 39.4 | 47.7 | 54.7 | 56.0 | 47.4 | 38.8 | 29.7 | 20.5 |
| DBG Lin et al. (2020) AAAI | 37.3 | 46.7 | 54.5 | 57.8 | 49.4 | 39.8 | 30.2 | 21.7 |
| BC-GNN Bai et al. (2020) ECCV | 40.5 | 49.6 | 56.3 | 57.1 | 49.1 | 40.4 | 31.2 | 23.1 |
| BSN++ Su et al. (2021) AAAI | 42.4 | 49.8 | **57.6** | 59.9 | 49.5 | 41.3 | 31.9 | 22.8 |
| RTD-Net Tan et al. (2021) ICCV | 41.1 | 49.0 | 56.1 | 53.9 | 48.9 | 42.0 | 33.9 | 23.4 |
| TCANet Qing et al. (2021b) CVPR | 42.1 | 50.5 | 57.1 | 60.6 | 53.2 | 44.6 | 36.8 | 26.7 |
| TSP Alwassel et al. (2021) ICCV | - | - | - | 69.1 | 63.3 | 53.5 | 40.4 | 26.0 |
| E2E-TAD Liu et al. (2022a) TIP | - | - | - | **69.4** | **64.3** | **56.0** | **46.4** | **34.9** |
| Ours | **42.8** | **50.9** | 57.5 | 56.7 | 51.2 | 43.2 | 36.5 | 26.8 |

Table 2: Comparison of proposal generation results using AR@AN and mAP on THUMOS14.

| Method | AR@AN | | mAP | | | |
|---|---|---|---|---|---|---|
| | @1(val) | @100(val) | 0.5 | 0.75 | 0.95 | Average |
| Lin et al. Lin et al. (2017) CVPRW | - | 73.0 | 44.4 | 29.7 | 7.1 | 29.2 |
| BSN Lin et al. (2018) ECCV | *34.3* | 76.5 | 46.5 | 30.0 | 8.0 | 30.0 |
| BMN Lin et al. (2019) ICCV | - | 75.0 | 50.1 | 34.8 | 8.3 | 33.9 |
| SSN Zhao et al. (2020) IJCV | 32.2 | 74.2 | 39.1 | 23.5 | 5.5 | 24.0 |
| BSN++ Su et al. (2021) AAAI | *34.3* | 76.5 | 51.3 | *35.7* | 8.3 | 34.9 |
| RTD-Net Tan et al. (2021) ICCV | 33.1 | 73.2 | 46.4 | 30.5 | *8.6* | 30.5 |
| TCANet Qing et al. (2021b) CVPR | 34.6 | *76.1* | **51.9** | 34.9 | 7.5 | 34.4 |
| TSP Alwassel et al. (2021) ICCV | **34.9** | **76.6** | 51.3 | **37.1** | 9.3 | **35.8** |
| E2E-TAD Liu et al. (2022a) TIP | - | - | *50.5* | *36.0* | **10.8** | 37.1 |
| Ours | 34.6 | 75.2 | *50.5* | 34.3 | 8.8 | 34.8 |

Table 3: Analogous results for the ActivityNet Dataset

| Method | AR@AN | | | mAP | | |
|---|---|---|---|---|---|---|
| | @50 | @100 | @200 | 0.1 | 0.3 | 0.5 |
| JCR-RNN Li et al. (2016) ECCV | - | - | - | 49.9 | 44.0 | 34.5 |
| TAP-B-M Song et al. (2018) TIP | - | - | - | 51.3 | 48.0 | 35.2 |
| RTD-Net Tan et al. (2021) ICCV | 43.6 | 53.5 | 62.1 | 67.4 | 63.2 | 58.9 |
| Ours | **44.1** | **55.4** | **63.2** | **68.1** | **64.1** | **59.3** |

Table 4: Analogous results for the PKU-MMD Dataset Chunhui et al. (2017).

To evaluate the quality of the generated proposals, we calculate AR@AN and mAP on THUMOS14, ActivityNet and PKU-MMD, respectively. Tables 2, 3 and 4 show AR@AN and mAP under different tIoU thresholds. For a fair comparison, we retrained Tan et al. (2021) on the PKU-MMD dataset. The AR@AN and mAP scores of the other state-of-the-art methods are from their papers. Our MatNet superior results on THUMOS14 and PKU-MMD, and achieves comparable performance on ActivityNet. Specifically, on the PKU-MMD dataset, MatNet outperform the rest methods, implying that MatNet works best on the indoor dataset, where poses are clearer without too much noise. In addition, on THUMOS14, MatNet achieves comparable results on the mAP scores compared to the state-of-the-art methods under high tIoU, indicating the proposals generated by MatNet have more precise boundaries, and are robust to occlusion and multi-person scenarios. Moreover, although the annotations of ActivityNet are sparse (about 1.41 activity instances per video), MatNet still can achieve comparable results. Finally, our method achieves AR@AN and mAP improvements over our baseline Tan et al. (2021) at all AN and tIoU thresholds by a big margin, demonstrating that instead of using a pre-trained model to generate the boundary score, using a boundary transformer to generate boundary score and training jointly can boost the performance.

## 5.3 ABLATION STUDY

| Method | AR@AN | | | mAP | | | | |
|---|---|---|---|---|---|---|---|---|
| | @50 | @100 | @200 | 0.3 | 0.4 | 0.5 | 0.6 | 0.7 |
| Baseline | 33.2 | 42.1 | 49.3 | 47.1 | 42.8 | 38.2 | 25.3 | 18.3 |
| + P | 40.4 | 48.4 | 56.8 | 55.3 | 49.7 | 42.1 | 35.7 | 25.8 |
| + P + c | 42.6 | 49.2 | 57.1 | 56.5 | 50.8 | 43.0 | 36.3 | 26.3 |
| + P + c + m | **42.8** | **50.9** | **57.5** | **56.7** | **51.2** | **43.2** | **36.5** | **26.8** |

Table 5: Ablation study of different combinations of the components in MatNet using AR@AN and mAP on THUMOS14.

| Method | AR@AN | | mAP | | | |
|---|---|---|---|---|---|---|
| | @1(val) | @100(val) | 0.5 | 0.75 | 0.95 | Average |
| Baseline | 30.1 | 71.8 | 43.7 | 29.8 | 7.6 | 29.2 |
| + P | 33.4 | 74.3 | 49.3 | 33.8 | 8.1 | 33.8 |
| + P + c | 34.1 | 74.9 | 50.1 | 34.1 | 8.5 | 34.3 |
| + P + c + m | **34.6** | **75.2** | **50.5** | **34.3** | **8.8** | **34.8** |

Table 6: Analogous results for the ActivityNet.

| Method | AR@AN | | | mAP | | |
|---|---|---|---|---|---|---|
| | @50 | @100 | @200 | 0.1 | 0.3 | 0.5 |
| Baseline | 40.0 | 51.1 | 58.5 | 64.1 | 58.3 | 55.8 |
| + P | 42.1 | 53.5 | 60.9 | 66.4 | 61.1 | 57.6 |
| + P + c | 43.5 | 55.1 | 62.6 | 67.5 | 63.7 | 58.7 |
| + P + c + m | **44.1** | **55.4** | **63.2** | **68.1** | **64.1** | **59.3** |

Table 7: Analogous results for PKU-MMD Chunhui et al. (2017).

Tables 5, 6 and 7 show the results of ablation studies on the effectiveness of 3D pose representation **P**, atomic movement representation **m** and semantic movement representation **c** on all the datasets, measured by AR@AN and mAP. The baseline only takes I3D **x** as input. The results show that using 3D pose improves the performance significantly, which may be because the pose is helping in differentiating the frames containing human movement from the background frames not containing any human. When **P**, **c** and **m** are added cumulatively, the performance further improves steadily.

## 5.4 QUALITATIVE VISUALIZATIONS

Figure 2 shows the visualization of two sample results. During inference, we use bipartite matching as in Tan et al. (2021) to select the top-1 proposals from all the generated candidate proposals, which is order-independent. The blue color in semantic movement features means there are recognized body part movements, and the high values of atomic movement features also represent large human movement. We can see that the generated proposals are well aligned with these input features.

## 6 LIMITATIONS AND FUTURE WORK

Our performance will suffer when pose information is missing, e.g., in many everyday videos containing complex human activities such as applying makeup, where only face is visible and the activities involve subtle (e.g., finger or within face) movements, and surfing, where the person image has a small number of pixels. In furture, we plan to consider action appropriate, additional modalities to enhance performance for the actions involved, such as audio.

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

# Appendices

## A  CODE

We have made our training and inference code available to the reviewers in the submission zip file.

## B  DEMO VIDEOS

The evaluation we present in the main paper Sec. 5 Fig. 2 is in terms of statistics of matches provided by our proposal generation method with the ground truth. We evaluate the generated proposals in terms of their match with the ground truth number and locations of actions detected, and their starting and ending frames.

Here we present some representative video clips identified by our method within a larger range of actions. This shows the correspondence of our generated proposals with the sequence of frames associated with the proposals. We mark each frame of a test video with all the generated and ground-truth action proposals. Viewing these videos can bring out the cases where non-existing actions are incorrectly detected and existing actions are missed.

The demo videos we use are selected from two sources.

First, from the public available datasets - Thumos14, ActivityNet and PKU-MMD - used in the paper.

Second, we use videos taken from outside these datasets, to evaluate the Out-of-Distribution performance of our method. These demo videos can be found in the submission zip file. All of these video clips contain at least one of the following three scenarios: (1) Clips with no humans in them, (2) Clips with humans without movement, and (3) Clips with moving humans.

From the results shown on these videos, and corresponding to the statistics of results in the main paper, we see that: (1) Our model can identify the clips with no humans in them. This is a result due to our human movement descriptors. In addition, (2) we also detect and skip the clips where the person is stationary, which is very likely due to the integration of human detection with the use of our semantic descriptors of human movement (which help distinguish between human movement, vs. still human+background movement). Finally, (3) we are able to detect and segment the clips with human action, and distinguish the actions therein. Although we do not aim at recognizing specifically what action it is, our model can distinguish between different human actions, and between human and background movements.

