# OpenReview forum: "Movement-to-Action Transformer Networks for Temporal Action Proposal Generation"
_ICLR.cc/2023/Conference — Submitted to ICLR 2023_

### Official Review · Reviewer_2bvy · 2022-10-23

**Confidence:** 4
**Correctness:** 2
**Technical Novelty And Significance:** 2
**Empirical Novelty And Significance:** 2
**Recommendation:** 3

**Clarity, Quality, Novelty And Reproducibility:**

Please mainly see the weaknesses sections for details, my concern mains lies in related works and the neccessity of LMA.

**Strength And Weaknesses:**

This paper proposes a new action proposal generation method based on Laban Movement Analysis. Based on LMA, the author proposes Transformer based methods to generate action proposals.

I think this paper is somehow novel, which includes new action representations. However, I think this paper has following drawbacks:

1. The author misses recent state-of-the-art literature and comparisons. The author claimed that they achieves state-of-the-art performance over three benchmark datasets: THUMOS14, ActivityNet v1.3 and PKU-MMD. However, the most recent literature in the Table is TSP from ICCV 2021. In AAAI/CVPR 2022/recent journals, there are several paper including TadTR [1], E2E-TAD [2], ActionFormer [3] and CPNet [4]. Also, TadTr, E2E-TAD and ActionFormer are Transformer based methods for temporal action localization. Missing discussion with recent state-of-the-art models are a fatal drawback of this paper.

2. The author claims to use Laban Movement Analysis to generate action proposals. However, I'm a little bit confused: LMA needs extra 3D poses (though they directly extract the pose from the original video using LCRNet). Extra information does not bring much performance gain to the model compared to existing baselines. However, existing baselines does not use extra 3D poses, they only use RGB/Flow features. Thus the importance of LMA can be neglected.

1: Liu X, Wang Q, Hu Y, et al. End-to-end temporal action detection with transformer[J]. IEEE Transactions on Image Processing, 2022, 31: 5427-5441.

2: Liu X, Bai S, Bai X. An Empirical Study of End-to-End Temporal Action Detection[C]//Proceedings of the IEEE/CVF Conference on Computer Vision and Pattern Recognition. 2022: 20010-20019.

3. Zhang C, Wu J, Li Y. Actionformer: Localizing moments of actions with transformers[J]. arXiv preprint arXiv:2202.07925, 2022.

4: Hsieh H Y, Chen D J, Liu T L. Contextual Proposal Network for Action Localization[C]//Proceedings of the IEEE/CVF Winter Conference on Applications of Computer Vision. 2022: 2129-2138.

**Summary Of The Paper:**

In this paper, the author proposes a new method, which utilizes RGB/Flow/3D poses to compute LMA-based low-to-high-level movement features, and uses Transformer to generate action proposals. The proposed method achieves "state-of-the-art" performance on major benchmark datasets including THUMOS14, ActivityNet and the newly PKU-MMD dataset.

**Summary Of The Review:**

In general, my review lies in the reject side. Cause the paper does not comparison to SoTA and the LMA related/performance issue, I would give this paper a reject rating.

---

> ### Author Response · Authors · 2022-11-19
> **Response to Reviewer 4 (2bvy)**
>
> We would like to thank the reviewer for their helpful comments. We are glad that the reviewer acknowledges the novelty of using LMA as new action representations.
>
> First, we totally agree that we can improve the submission by adding comparisons with more SoTA methods. We have added the following SoTA methods [1-10] in the related work section and experiment section.
>
> Secondly, we thank the reviewer for the insightful observation that the performance of the model with extra pose-related information is not significantly higher than the model using raw features only. We would like to clarify that in the ablation study, using pose-related information (adding p+c+m) significantly improves our baseline (I3D) by 46% and 19% on the Thumos14 and ANet, respectively. We think that it will be promising to embed our pose-related modules into other more powerful baselines.
> We hope these changes and clarification will encourage you to change your score.
>
> [1] G.Che ́ron, I.Laptev, and C.Schmid. P-CNN: Pose-based cnn features for action recognition. In ICCV, 2015.
>
> [2] C. Gu, C. Sun, S. Vijayanarasimhan, C. Pantofaru, D. A. Ross, G. Toderici, Y. Li, S. Ricco, R. Sukthankar, C. Schmid, et al. Ava: A video dataset of spatio-temporally localized atomic visual actions. arXiv 2017.
>
> [3] M. Zolfaghari, G. L. Oliveira, N. Sedaghat, and T. Brox. Chained multi-stream networks exploiting pose, motion, and appearance for action classification and detection. In ICCV 2017.
>
> [4] V. Choutas, P. Weinzaepfel, J. Revaud, and C. Schmid. Potion: Pose motion representation for action recognition. In CVPR 2018
>
> [5] D. Zhang, G. Guo, D. Huang, and J. Han. Poseflow: A deep motion representation for understanding human behaviors in videos. In CVPR 2018.
>
> [6] Asghari-Esfeden, S., Sznaier, M. and Camps, O. Dynamic motion representation for human action recognition. WACV 2020.
>
> [7] Liu X, Wang Q, Hu Y, et al. End-to-end temporal action detection with transformer. IEEE TIP, 2022
>
> [8] Liu X, Bai S, Bai X. An Empirical Study of End-to-End Temporal Action Detection, In CVPR 2022
>
> [9] Hsieh H Y, Chen D J, Liu T L. Contextual Proposal Network for Action Localization, In CVPR 2022
>
> [10] Zhang C, Wu J, Li Y. Actionformer: Localizing moments of actions with transformers. arXiv 2022

---

### Official Review · Reviewer_Exnj · 2022-10-24

**Confidence:** 5
**Correctness:** 3
**Technical Novelty And Significance:** 2
**Empirical Novelty And Significance:** 2
**Recommendation:** 3

**Clarity, Quality, Novelty And Reproducibility:**

Using LMA as a new action descriptor to generate action proposals is novel but the overall technical contribution is not very significant.

**Strength And Weaknesses:**

Strengths:
1.	It is new to see that Laban Movement Analysis (LMA) can be used as a new action descriptor for representing actions and can be exploited to generate action proposals. Such a design is novel in the TAPG task.
2.	The proposed method achieves comparable results compared to SOTA methods.
Weaknesses&Questions:
Though the overall paper is interesting, I have the following concerns:
1.	The technical contribution is not very significant. To my understanding, the transformer-based TAPG method has been studied in previous works such as RTD-Net (ICCV2021). The main difference between RTD-Net and the proposed method lies in the visual features. This paper additionally considers 3D pose and the action features obtained by Laban Movement Analysis (LMA). However, LMA has been proven to be effective in dance action recognition, it is not that surprising to see that the use of additional features could improve the performance. Thus, such a choice may not bring enough insights to the community.
2.	Tied with the above question, if LMA features are indeed useful for representing action, why not use LMA features to replace the action features in other TAPG or TAD methods? How will the performance change when only LMA features are used in the proposed framework?
3.	Could the LMA features (compared with I3D features) make the proposed more generalizable?
4.	From Figure1, it seems that the proposed method could be affected by the occlusion problem since the detector may fail to detect precise pose. Is the proposed method sensitive to the pose detectors?
5.	From Section 4.1, it seems that the proposed method requires manual annotations for the timestamps.
6.	As for the proposal transformer, the authors claim that they do not need an additional backbone to generate and save boundary scores (compared with RTD-Net). Does this design bring improvements?
7.	From Table2, it is confusing that the proposed method achieves good scores in terms of AR@AN (with more high-quality proposals) but obtains worse results in terms of mAP (worse detection performance).

**Summary Of The Paper:**

In this paper, the authors propose a transformer-based method for the temporal action proposal generation task. Compared with existing methods, the main difference lies in the choice of visual features, which are obtained from the Laban Movement Analysis (LMA) method that are originally designed for action analysis. The proposed method achieves comparable results compared to SOTA methods.

**Summary Of The Review:**

To my understanding, the idea of using LMA to represent actions for the TAPG task is new and interesting. However, considering the limited technical contribution (see detailed comments in the Weakness Section), I vote for rejecting this paper for its current shape. One direction that may improve the paper is testing LMA features for other action recognition or detection methods.

---

> ### Author Response · Authors · 2022-11-19
> **Response to Reviewer 3 (Exnj)**
>
> We would like to thank the reviewer for their constructive and detailed comments and for taking the time to evaluate our work. We are glad that the reviewer finds our LMA feature as an action descriptor novel and acknowledges that our model’s performance is comparable to SoTA.
>
> Firstly, with regard to the contribution, we agree with the reviewer that LMA has been used in dance action recognition and transformers have been used in the TAPG task. However, we would like to clarify that, firstly, LMA is a concept designed by dance experts, and there is no attempt to apply it to general daily-life videos, which are challenging for occlusions, a large variety of activities, low-resolution, and moving cameras. On the contrary, our method is robust to occlusions, evaluated on daily activities, and explicitly defines the expression of LMA, making it easy to use. Secondly, many transformer-based methods such as RTD-Net rely on a pre-trained backbone to extract boundary scores, while our end-to-end design is easy to adapt to new datasets and can properly fuse a range of low (atomic) to high (semantic) level human movements for TAPG.
>
> Second, we understand that the reviewer’s primary concern is about the effectiveness and generalization of the LMA feature. We want to highlight that we conducted an ablation study on three different datasets in this work to show the benefits of adding the LMA feature (denoted as semantic movement representation c in Table 5-7). As adding the LMA feature achieves competitive results on all datasets, we believe the LMA feature is highly generalizable. We agree with the reviewer that using LMA features to replace the features in other methods can further show the power of LMA. However, as we stated in the contribution section, while LMA as semantic features can quantitatively describe human movement at high level, the atomic features as basic primitives can describe the movements of each body part in low-level. The combination and fusion of both features can provide a range of low (atomic) to high (semantic) level human movement features. In our model, we train multiple MLP layers to learn the fusion of these features, which sadly needs additional modifications if tested on other methods.
>
> We see that the reviewer raises an interesting question about the benefits of the end-to-end design of our architecture, compared to many methods such as RTD-Net relying on using the BSN model as the backbone to generate the TEM score as boundary score. First, the BSN backbone has to be trained separately for different datasets, while our end-to-end design allows us to easily adapt to different datasets. Second, the BSN is written in TensorFlow while most SoTA methods are written in Pytorch, which brings difficulty in deploying on devices. Lastly, our baseline method in Ablation Study Table 5-6 is exactly the same as the RTD-Net except that we replace the pre-trained BSN backbone with a boundary transformer. The performance of the baseline is still comparable to the RTD-Net which uses a powerful pre-trained backbone.
>
> Finally, we would like to answer the reviewer’s remaining questions. Regarding that our method could be affected by occlusion problems causing the failure of the pose detector, we want to highlight that our method is robust to occlusions by other humans or objects because our underlying human pose detector (using LCRNet) has such robustness. Regarding the manual annotation for the timestamps, our method will automatically estimate the timestamps, and we save the timestamps as atomic features for more efficient training of MatNet. We will update our draft to avoid this confusion. Regarding the good average recall but low mAP, our model achieves the best mAP when tIoU is high, meaning that we have more precise boundaries.
>
> We thank you again for your precious time in reviewing our paper and providing valuable comments. We hope our clarification of the contributions will encourage you to change your opinion.

---

### Official Review · Reviewer_dxzc · 2022-10-24

**Confidence:** 3
**Clarity, Quality, Novelty And Reproducibility:** 1. Clarity is good, but link between …
**Correctness:** 3
**Technical Novelty And Significance:** 3
**Empirical Novelty And Significance:** 2
**Recommendation:** 6

**Strength And Weaknesses:**

Strengths
1. Paper is well-written and has good theorical motivation.
2. The experimental results somewhat support the main claims of the paper: LBM framework is useful for solving the TAPG problem.
3. Sufficient ablation experimental results were done to demonstrate the contributions of the different modules/subnets.

Weaknesses
1. The direct link from LBM to the TAPG is weak. While LBM can be used to describe an action, it is not clear how the measures are directly translated to solving the TAPG problem.
2. While the experimental results are promising, they are quite mixed. The exceptional results for PKU-MMD is somewhat weakened by the fact that only 3 other methods were compared.

**Summary Of The Paper:**

Paper proposed a novel approach of using end-to-end Transformer network to compute 8 semantic measures based on Laban Movement Analysis (LMA) of the input video to solve the Temporal Action Proposal Generation problem.

The proposed network shows comparable results against SOTA methods on 3 datasets, THUMBOS14, ActivityNet and PKU-MMD.

**Summary Of The Review:**

This is a very interesting and somewhat novel paper. It draws from established framework to solve a challenging CV problem. However, the link between the two were not clearly explained and established in the paper. Furthermore, the mixed experimental results also throw the question wide-open if the proposed approach is indeed as useful as the paper is claiming.

---

> ### Author Response · Authors · 2022-11-19
> **Response to Reviewer 2 (dxzc)**
>
> We appreciate the reviewer providing useful feedback and acknowledge our paper is well-written and novel, and experimental results are supportive of our claims.
>
> Firstly, we see that the reviewer is concerned about the lack of clarity about the translation of the LBM to TAPG task. We note that Section 3 and Table 1 describe each factor of LBM that can capture the position, direction, rotation, velocity, acceleration, distance, curvature and volume associated with the movement. We use these Factors as bases to obtain temporal action proposals through MatNet. MatNet automatically determines combinations of the Factors most suited for action detection and localization. In addition, we use TAPG as a downstream task to evaluate the LBM features’ capability to extract action information, and in the future, we will include more downstream tasks such as action recognition. We will update our draft to highlight how MatNet combines the LBM factors to generate action proposals.
>
> We agree that our paper will be better if we can have more comparison methods on the PKU-MMD dataset. However, most of the TAPG works are evaluated on the popular Thumos14 and ANet datasets containing in-the-wild videos. We add additional experiments on the in-door camera-fixed PKU-MMD dataset, and hope to give a taste to readers about the generalization capability of our model on different scenarios.

---

### Official Review · Reviewer_R3np · 2022-10-25

**Confidence:** 4
**Correctness:** 4
**Technical Novelty And Significance:** 4
**Empirical Novelty And Significance:** 3
**Recommendation:** 8

**Clarity, Quality, Novelty And Reproducibility:**

The paper is clear and understandable. The idea is new and seems to be outperforming SOTA.

**Strength And Weaknesses:**

- The method is robust to occlusions
- the boundary and proposal networks can be trained end2end jointly.
Weaknesses:
- this idea is heavily based on human body movements, while a lot of actions might include minimal body movements, or the person acting could be heavily covered and therefore the body movement analysis and pose estimation could include a lot of error.


**Summary Of The Paper:**

This paper focuses on using human movements for action proposal generation. The proposed MAT (end2end Movement to action transformer networks) use a range of low to high level human movements for temporal action proposal generation.

**Summary Of The Review:**

I think the proposed idea has some novelty in it where there are two movement descriptors (atomic and semantic). I would be interested in seeing the computational complexity of this algorithm as well as its performance on newer datasets.
I would also include comparisons with more SOTA methods and cite them, some examples here:
- V. Choutas, P. Weinzaepfel, J. Revaud, and C. Schmid. Potion: Pose motion representation for action recognition. In CVPR 2018, 2018.
- C. Gu, C. Sun, S. Vijayanarasimhan, C. Pantofaru, D. A. Ross, G. Toderici, Y. Li, S. Ricco, R. Sukthankar, C. Schmid, et al. Ava: A video dataset of spatio-temporally localized atomic visual actions. arXiv preprint arXiv:1705.08421, 3(4):6, 2017.
- G.Che ́ron,I.Laptev,andC.Schmid.P-cnn:Pose-basedcnn features for action recognition. In Proceedings of the IEEE international conference on computer vision, pages 3218– 3226, 2015.
- Asghari-Esfeden, S., Sznaier, M. and Camps, O., 2020. Dynamic motion representation for human action recognition. In Proceedings of the IEEE/CVF Winter Conference on Applications of Computer Vision (pp. 557-566).
- D. Zhang, G. Guo, D. Huang, and J. Han. Poseflow: A deep motion representation for understanding human behaviors in videos. In Proceedings of the IEEE Conference on Computer Vision and Pattern Recognition, pages 6762–6770, 2018.
- M. Zolfaghari, G. L. Oliveira, N. Sedaghat, and T. Brox. Chained multi-stream networks exploiting pose, motion, and appearance for action classification and detection. In Com- puter Vision (ICCV), 2017 IEEE International Conference on, pages 2923–2932. IEEE, 2017.

---

> ### Author Response · Authors · 2022-11-19
> **Response to Reviewer 1 (R3np)**
>
> Thank you for your helpful comments. We are glad that you find our method is robust to occlusions and can be trained end-to-end jointly.
>
> First, we totally agree that we can improve the submission by adding comparisons with more SoTA methods. We have added the following SoTA methods in the related work section and experiment section.
>
> Second, regarding the time analysis in inference (the experiment of inference speed is conducted on a single RTX 1080Ti GPU): Our inference time per sample is 4.2s per vs 3.3 s per sample for RTD-Net. Compared with RTD-Net which directly uses pre-stored boundary scores produced by BSN, we simultaneously predict the boundary score and generate action proposals. Regarding memory, our method (two transformers) uses 1,991 MB GPU while RTD-Net (one transformer) uses 1,551 MB.
>
> [1] G.Che ́ron, I.Laptev, and C.Schmid. P-CNN: Pose-based cnn features for action recognition. In ICCV, 2015.
>
> [2] C. Gu, C. Sun, S. Vijayanarasimhan, C. Pantofaru, D. A. Ross, G. Toderici, Y. Li, S. Ricco, R. Sukthankar, C. Schmid, et al. Ava: A video dataset of spatio-temporally localized atomic visual actions. arXiv 2017.
>
> [3] M. Zolfaghari, G. L. Oliveira, N. Sedaghat, and T. Brox. Chained multi-stream networks exploiting pose, motion, and appearance for action classification and detection. In ICCV 2017.
>
> [4] V. Choutas, P. Weinzaepfel, J. Revaud, and C. Schmid. Potion: Pose motion representation for action recognition. In CVPR 2018
>
> [5] D. Zhang, G. Guo, D. Huang, and J. Han. Poseflow: A deep motion representation for understanding human behaviors in videos. In CVPR 2018.
>
> [6] Asghari-Esfeden, S., Sznaier, M. and Camps, O. Dynamic motion representation for human action recognition. WACV 2020.
>
> [7] Liu X, Wang Q, Hu Y, et al. End-to-end temporal action detection with transformer. IEEE TIP, 2022
>
> [8] Liu X, Bai S, Bai X. An Empirical Study of End-to-End Temporal Action Detection, In CVPR 2022
>
> [9] Hsieh H Y, Chen D J, Liu T L. Contextual Proposal Network for Action Localization, In CVPR 2022
>
> [10] Zhang C, Wu J, Li Y. Actionformer: Localizing moments of actions with transformers. arXiv 2022.

---

### Author Response · Authors · 2022-11-19
**Response to All Reviewers**

We gratefully acknowledge and thank the reviewers’ insightful comments. We are glad that all four reviewers agree on the novelty of our method, where we use LMA as a new action descriptor for representing actions. In addition, we also appreciate that Reviewer#2 (dxzc) and Reviewer#3 (Exnj) find our paper with sufficient experimental and ablation study results to support our claims, and our results are comparable to the SOTA.

As we have elaborated in each individual response, if given more time and space, we can address all the reviewers’ comments with additional experiments and improve the draft.

---

### Decision · Program_Chairs · 2023-01-20

**Decision:**

Reject

**Justification For Why Not Higher Score:**

1. Unconving experiments
2.  Limited technical contribution

**Justification For Why Not Lower Score:**

NA

**Metareview: Summary, Strengths And Weaknesses:**

This paper was reviewed by four experts in the field and received a mixed score. The main concerns are unconvincing experiments and lack of clarity. The authors did a good job of rebuttal and addressed many of the concerns. However, AC still feels that more work is needed to get it to the best version. AC also agrees that this work can be much stronger with additional experiments. While this paper clearly has merit, the decision is not to recommend acceptance. The authors are encouraged to consider the reviewers' comments when revising the paper for submission elsewhere.